# Perioperative Techniques for the Use of Botulinum Toxin in Overactive Bladder: Results of a Multinational Online Survey of Urogynecologists in Germany, Austria, and Switzerland ^[note 1]^

**DOI:** 10.3390/jcm12041462

**Published:** 2023-02-12

**Authors:** Sören Lange, Marianne Koch, Rainer Lange, Heinrich Husslein, Wolfgang Umek, Barbara Bodner-Adler

**Affiliations:** 1Department of General Gynecology and Gynecologic Oncology, Medical University of Vienna, 1090 Vienna, Austria; 2DieGyn-Praxis, Lampertheim/Mainz/Bad Kreuznach, 55232 Alzey, Germany

**Keywords:** botulinum toxin, overactive bladder, survey

## Abstract

Introduction and Hypothesis: Botulinum toxin (BoNT) is a widely used treatment for overactive bladder (OAB). Despite its common use, no standardized treatment regimen exists so far. The aim of this survey was to evaluate the variation in perioperative treatment strategies among members of the German-speaking urogynecologic societies. Materials and Methods: A clinical practice online survey was carried out between May 2021 and May 2022, and all members of the German, Swiss, and Austrian urogynecologic societies were invited to participate. Participants were grouped in two ways. First, they were grouped into (1) urogynecologists with board certification and (2) non-board-certified general obstetricians and gynecologists (OBGYNs). Second, we set a cut-off at 20 transurethral BoNT procedures per year to differentiate between (1) high- and (2) low-volume surgeons. Results: One hundred and six completed questionnaires were received. Our results demonstrated that BoNT is mostly used as a third-line treatment (93%, *n* = 98/106), while high-volume surgeons used it significantly more often as a first/second-line treatment (21% vs. 6%, *p* = 0.029). Large variations existed in the use of perioperative antibiotics, preferred sites of injection, the number of injections, and the timing of the measurement of the postvoid residual volume (PVRV). Forty percent of participants did not offer outpatient treatment to patients. Local anesthesia (LA) was mostly used by board-certified urogynecologists (49% vs. 10%, *p* < 0.001) and high-volume surgeons (58% vs. 27%, *p* = 0.002). Injections into the trigone were also more often performed by board-certified urogynecologists and high-volume surgeons (22% vs. 3% (*p* = 0.023) and 35% vs. 6% (*p* < 0.001), respectively). PVRV was controlled between weeks 1 and 4 by only 54% of participants (*n* = 57/106). Clean intermittent self-catheterization (CISC) was infrequently taught (26%). Conclusions: Our survey confirmed that BoNT is widely used by urogynecologists in the three German-speaking countries, but practice patterns vary widely, and no standardized method could be detected, despite interviewing urogynecologic experts. These results clearly demonstrate that there is a need for studies to define standardized treatment strategies for the best perioperative and surgical approach regarding the use of BoNT in patients with OAB.

## 1. Introduction

Botulinum toxin (BoNT) is a widely used treatment for refractory idiopathic overactive bladder (OAB) [1]. Botulinum toxin is considered one of the most potent neurotoxins, but low-dose applications into the vesical detrusor muscle are considered a safe treatment for OAB [2,3,4]. Via the proteolysis of SNAP-25, botulinum toxin decreases the secretion of acetylcholine into the neuronal synapse, causing a chemodenervation of afferent and efferent nerves [5]. It has a better efficacy than oral antimuscarinic drugs to cure urge urinary incontinence (UUI) [6]. Compared to sacral neuromodulation, BoNT has shown a similar efficacy in decreasing UUI episodes, but patients show higher satisfaction with BoNT after 24 months [7].

BoNT is recommended as a third-line treatment when behavioral and pharmacologic treatments fail [8,9,10]. Nevertheless, no standardized perioperative regimen exists so far, and relevant specifications, such as the frequency of application, the number of injections, and the administration of perioperative antibiotics, remain unclear [5,11]. It is also unknown how many urogynecologists use BoNT to treat OAB. In Austria, Switzerland, and Germany, the treatment is reimbursed or paid directly by the mandatory health insurances. Therefore, theoretically, all eligible patients should have relatively easy access to it, and numerous urogynecologists should offer it.

We aimed to conduct a study to evaluate the patterns of the use of BoNT in overactive bladder among German-speaking urogynecologists and compare practices between board-certified and non-board-certified urogynecologists.

## 2. Materials and Methods

A clinical practice online survey about the use of botulinum toxin for OAB was carried out between May 2021 and May 2022. All members of the German, Swiss, and Austrian Urogynecologic Societies received an e-mail explaining the research question and asking for their cooperation in an internet-based survey. A link in the e-mail directed them to a website containing an internet-based questionnaire.

The survey included 32 questions covering the following topics: indications for treatment, the context of treatment, the type of product and dosage, the type of anesthesia, the technique of application, perioperative management, and professional and treatment experience (see Appendix A). The questionnaire consisted of multiple-choice questions with either single-answer or multiple-answer options. Eight questions gave the possibility for open answers (the indication of treatment, the type of product used, the type of anesthesia used by anesthesiologist, the type of cystoscope used, the preferred site of injection, the timing of the measurement of the postvoid volume, the timing of treatment repetition, and the type of institution). The survey took approximately 5 min for each participant to complete.

Before launching the study, three urogynecologists were consulted, and based on their recommendations, changes were made to the questions and answers. Participation in the survey was anonymous, and no financial compensation was given to participants. Every participant was only allowed to answer the questionnaire once.

### 2.1. Participants

We asked the urogynecologic working groups in Germany (AGUB), Switzerland (AUG), and Austria (AUB) for permission to contact their members. After receiving permission, all members of the three working groups were invited to participate via a link sent by mail. Each urogynecologic working group had different bylaws by which it was allowed to contact the members: AGUB allowed only indirect contact via a newsletter, AUG allowed personal contact, and AUB permitted personal contact via the secretary of the association but not by the researchers. None of the working groups had a list of active practicing members.

Participants were grouped in two ways. First, they were grouped into urogynecologists with and without board certification: (1) board-certified urogynecologists and (2) non-board-certified general obstetricians and gynecologists (OBGYNs). Second, we set a cut-off at 20 transurethral BoNT procedures per year to differentiate between (1) high- and (2) low-volume surgeons.

### 2.2. Statistical Analysis

Descriptive statistics were calculated using SPSS Statistics for Mac, version 27.0 (SPSS Inc., Chicago, IL, USA). Proportions were calculated for ordinal variables. A chi-squared test was used to assess differences between the groups. Statistical significance was defined as *p* < 0.05. Incomplete surveys were excluded.

### 2.3. Ethical Considerations

The survey did not contain any patient data. Therefore, in accordance with the Declaration of Helsinki, it was exempt from approval by the institutional review board.

## 3. Results

A total of 122 participants responded to the online survey, with 106 filling it out completely (87%). In total, 93% (*n* = 99/106) of respondents considered botulinum toxin as a treatment for overactive bladder, but only 86% (*n* = 91/106) of these performed the procedure themselves. The basic characteristics of the participants can be seen in Table 1. 

Thirty-seven percent of respondents (*n* = 39/106) declared themselves to be high-volume surgeons with over 20 transurethral BoNT procedures per year (see Table 2). Those who were board-certified urogynecologists were high-volume surgeons more often than non-board-certified gynecologists (45% vs. 14%, *p* = 0.003).

### 3.1. Treatment Indications

In total, 91% (*n* = 96/106) used BoNT to treat idiopathic OAB and 88% (*n* = 93/106) used BoNT to treat mixed urinary incontinence (MUI) with a predominant OAB, while 67% (*n* = 71/106) used it to treat neurogenic OAB. 

BoNT was mostly considered as a treatment in the case of failure of oral drug therapy or as a third-line therapy (93%, *n* = 98/106). Eleven percent (*n* = 12/106) also used BoNT as a first- or second-line treatment prior to oral drug therapy. After a failure of neuromodulation, 27% (*n* = 29/106) used BoNT as a treatment option. We did not find differences between the participants who performed the procedure themselves and those who referred patients to colleagues to receive the treatment or differences due to the line of treatment in which BoNT was used. On the other hand, high-volume surgeons used BoNT more often as a first-line treatment (21% vs. 6%, *p* = 0.029), even before oral drug therapy.

### 3.2. Technique of BoNT Application

In total, 60% proposed an outpatient treatment (*n* = 64/106), with 88% (*n* = 93/106) performing the procedure in an operating room. A marked needle to visually control the depth of the injection was used by 66% of all survey participants (*n* = 70/100).

Half of all participants performed the procedure under general anesthesia (50%, *n* = 87/106; see Table 2), and local anesthesia was only used by 39% of participants (*n* = 41/106). While board-certified urogynecologists and high-volume surgeons used local anesthesia significantly more often (49% vs. 10% (*p* < 0.001) and 27% vs. 58% (*p* = 0.002), respectively), general OBGYNs more often used spinal anesthesia or no anesthesia at all (54% vs. 28% (*p* = 0.019) and 14% vs. 3% (*p* = 0.026), respectively).

Perioperative antibiotics were used by 68% of participants (*n* = 72/106), with a significant difference between board-certified urogynecologists and non-board-certified gynecologists (71% vs. 59%, *p* = 0.015).

Almost all participants used Botox^®^ (99%, *n* = 105/106), with 88% applying 100 U (*n* = 93/106) and only a minority using other products. Seventy-two percent of participants applied the recommended number of 11 to 20 injections (*n* = 76/106). Only 17% (*n* = 18/106) applied a lower number of injections, while 11% (*n* = 12/106) used even more injections. No differences between the groups were found. 

Concerning the preferred sites of injection, significantly more high-volume surgeons and board-certified urogynecologists injected into the trigone (35% vs. 6% (*p* < 0.001) and 22% vs. 3% (*p* = 0.023), respectively), while most participants avoided injecting into this area (83%, *n* = 88/106). No differences between the groups were found for the other sites of injection.

### 3.3. Postoperative Care

Postoperative indwelling catheters were used by 48% (*n* = 51/106) of participants, and no difference was found between the groups. Clean intermittent self-catheterization was only taught perioperatively in a prophylactic manner by a minority of participants (26%, *n* = 27/106). 

The postvoid residual volume (PVRV) was measured by 81% of participants (*n* = 86/106), mostly prior to patient discharge (59%, *n* = 62/106; see Table 3). When not counting those participants who were screened during the first week and at one month to avoid counting doubles, a total of 57 participants (54%) screened for urinary retention during the recommended period between the first week and 1 month after the procedure. 

In the case of insufficient effect, 76% (*n* = 81/106) repeated the treatment, mostly between 4 and 6 months or between 7 and 12 months later (30% (*n* = 32/106) and 36% (*n* = 38/106), respectively). Nine percent (*n* = 9/106) of participants repeated the treatment in the first three months. Additionally, 2% (*n* = 2/106) only repeated the treatment at least 12 months after the last injection.

## 4. Discussion

Current guidelines propose a step-by-step approach to overactive bladder, with behavioral treatment as the first-line treatment, followed by oral drug therapy (anti-muscarinic or ß3-adrenoreceptor agonists) as second line [8]. Botulinum toxin is considered a third-line treatment as an alternative for neuromodulation. Nevertheless, a recent analysis from the UK found the use of onabotulinumtoxinA to be more cost-effective than oxybutynin when used as a second-line treatment [12]. In our survey, most participants used botulinum toxin after the failure of oral drug therapy. Interestingly, we found that high-volume surgeons used BoNT significantly more often as a first-line treatment prior to medical or conservative treatments compared to low-volume surgeons. The referral of a patient to another colleague to have the procedure performed did not influence the line of treatment. Even though the number of participants who did not perform the procedure themselves was rather low, we cannot conclude that participants are less willing to use the treatment if they must address patients to other centers to receive the treatment.

To this day, no study has investigated the effect of the number of interventions per surgeon per year on patient outcomes or surgical failure and complication rates regarding BoNT. Barba et al. investigated the learning curve in residents and only found improvements for operative time, subjective easiness, and perceived tolerability [13]. No improvements were found concerning the number of valid injections or complications. Unfortunately, this study included only nine patients per resident over a three-year period, which is much lower than most definitions for high-volume surgeons. It is therefore unclear how the learning curve develops when higher volumes are performed per surgeon. In a recent meta-analysis by Mowat et al. about differences in surgical outcomes for low- versus high-volume gynecologic surgeons in several gynecological surgery types, low-volume surgeons had more adverse outcomes and higher mortality [14]. In our cohort, only about a third of the participants in our survey indicated enough procedures per year to be classified as high-volume surgeons, though we have to acknowledge that our classification was arbitrarily chosen and could not be based on studies due to an absence of scientific data in this field. We found that board-certified urogynecologists significantly exceeded noncertified participants in the number of interventions performed per year. 

### 4.1. Type of Product and Dosage

Different types of BoNT products exist on the market. Problems with the different products arise from the fact that they are not bioidentical and have non-interchangeable potency units [15]. In the German-speaking countries, three products are used to treat overactive bladder, while only onabotulinumtoxinA has EMA approval for this indication. Consequently, more than 99% of participants used onabotulinumtoxinA as their preferred drug. 

One possible long-term complication of BoNT use can be the development of neutralizing antibodies, and this seems to differ between different formulations [16,17]. The possibly decreased efficacy of a certain formulation in the case of neutralizing antibodies can be clinically challenging and might necessitate a change in the product that is used [17]. It is therefore advisable that urogynecologists are trained in the application of different products. However, in our study only 7% of participants used two different types of formulations, while the rest only used onabotulinumtoxinA. A possible hindrance might be the fact that, with the exception of onabotulinumtoxinA, the use of other BoNT products for OAB is off-label in all three countries where the survey was conducted.

The approved dosage of onabotulinumtoxinA for the treatment of OAB is 100 U [10]. A recent secondary analysis of two multicenter randomized controlled trials assessing the treatment efficacy of onabotulinumtoxinA in women with OAB did not find more symptom improvements when higher doses were applied [18]. Catheterization and urinary tract infections were also not more common with the higher dosage (32% vs. 23%, aOR: 1.4, 95% CI: 0.8–2.4; 37% vs. 27%, aOR: 1.5, 95% CI: 0.9–2.6, respectively). In our survey, almost nine out of ten participants followed the current guidelines concerning the dosage of onabotulinumtoxinA [10].

### 4.2. Technique of Injection

Marked needles can be used to control the depth of injection, which is approx. 2 mm, to avoid bladder perforation. Still, Alsinnawi and colleagues found traces of possible extravesical extravasation in around 80% of cases in one MRI study [19]. In another study, Mehnert et al. showed contrast agent in the extraperitoneal fatty tissue lateral of the bladder dome in almost 18% of cases [20]. Both studies were small and therefore limited in their generalizability, but they indicate that extravesical extravasation is not infrequent. Fortunately, none of these studies found any adverse events in these patients. In our study, marked needles were only used by 66% of participants. We were not able to evaluate if other techniques to control the depth of injection (i.e., mixing the BoNT solution with dye or the sonographic measurement of the bladder wall thickness) were used.

Even though injecting botulinum toxin into the trigone was believed to cause vesico-ureteral reflux (VUR), no study has shown any increased risk [21]. On the other hand, patients who received trigone injections had more improved symptom scores when compared to patients without trigone injections, and recent guidelines have taken this into consideration [11,21]. However, a recent trial comparing trigonal-sparing versus trigonal-involved injections of onabotulinumtoxinA did not show that trigone injections are superior with regard to OAB symptoms at 3 months, but the incidences of voiding difficulties and UTIs were higher in the trigone injection group [22]. In our survey, we found that most surgeons still avoid the trigone, except for high-volume surgeons, who inject into this site significantly more frequently. Data are still not solid enough to recommend injections into the trigone, and we think that further studies should be conducted to help elucidate this detail.

A recent systematic review found a large variety in the number of injections used in different studies [23]. In our survey, 11–20 injections was the most commonly used regimen, which is in line with the market approval for onabotulinumtoxinA in most countries. Only non-board-certified participants applied more than 30 injections. 

### 4.3. Perioperative Antibiotics

Antibiotic treatment for intravesical procedures is still debated [8]. Additionally, women with urinary incontinence have higher rates of UTIs compared to the general population [24]. During the first 12 months after BoNT treatment, up to 35% of patients have UTIs, even when treated with perioperative antibiotics [7]. When comparing perioperative regiments, a three-day regimen of oral fluorchinolone was superior to a single dose of intramuscular ceftriaxone [25]. A treatment with ciprofloxacin started prior to injection decreased the risk of UTI when compared to a treatment started after injection [26]. On the other hand, when procedures are performed in patients who had a recent preoperative UTI but received prophylactic perioperative antibiotics during the intervention, no increase in postoperative UTIs could be found in a trial by Brickhaus and colleagues [27]. Consequently, certain authors recommend the use of perioperative antibiotics [28]. The European Association of Urology advocates that patients should be informed about the increased risk of UTIs [10]. In our study, we found that one third of participants did not use perioperative antibiotics. Moreover, board-certified urogynecologists used antibiotics significantly more often than others, even though the current German-speaking guidelines do not take a position on this topic [11]. Further studies should investigate this subject to give a clearer answer of which antibiotics might be beneficial. 

### 4.4. Type of Anesthesia

The procedure can be performed under different types of anesthesia [29,30]. To be cost-effective compared to sacral neuromodulation, local anesthesia should be the preferred technique [31]. In our survey, we found that in German-speaking countries general anesthesia is the most frequently used type of anesthesia. We were not able to evaluate the reason for this finding. Local anesthesia has very low risks in most patients while at the same time being well accepted [32]. No study was found comparing any type of anesthesia to placebo for BoNT, and it is therefore unknown if local anesthesia is superior to placebo.

### 4.5. Outpatient vs. Inpatient

The injection of BoNT is a short procedure with few short-term complications [32]. Urinary tract infections and urinary retention are the main intermediate complications [23]. Unfortunately, in the German-speaking countries, outpatient treatment is still quite underdeveloped, as we showed in our survey. A reason might be the way insurances reimburse for this treatment, but given the vast variety of different insurance systems in the German-speaking countries, no definite answer could be found. The use of local anesthesia instead of general anesthesia and no use of indwelling catheters after the procedure should permit patients to benefit from outpatient treatment [33].

### 4.6. Residual Volume

Botulinum toxin takes around 14 days to be effective [29]. Urinary retention that requires CISC occurs, in general, between 5 and 14 days after treatment in approx. 6% of patients [34]. In a prospective trial of more than 200 women who received 100 U of onabotulinumtoxinA, Miotla and colleagues found higher rates of CISC in women with three or more vaginal deliveries (OR 6.86, 95% CI 1.76–26.9, *p* < 0.01). Women with the need for CISC or with PVRVs of >200 mL were older than women with lower PVRVs in this study. Pelvic organ prolapse, body mass index, and comorbidities (i.e., hypertension, diabetes, and asthma) did not increase the risk of CISC or higher PVRV.

Therefore, all patients should be checked for increased postvoid volumes 7–14 days after treatment [11]. Our survey showed that only about half of respondents measured PVRV during this period.

Additionally, while CISC is a well-tolerated, and in most cases easy to learn, treatment for patients with urinary retention, guidelines diverge in recommending to teach or not to teach CISC prior to intervention [8,10,11]. For adult patients with neurogenic lower urinary tract dysfunctions, it is recommended to discuss the need for CISC before the treatment is started [35]. In our study, most participants gave no CISC instructions to patients prior to the intervention. Whether every patient needs to learn CISC prior to an intervention is up for debate, and there might be reasonable exceptions to treat patients with BoNT even if they are unable to perform CISC. Nevertheless, we think that most patients are capable of learning CISC and benefit from it in case urinary retention occurs.

### 4.7. Strengths

This was the first survey among urogynecologists investigating treatment patterns of the use of BoNT in overactive bladder. The multinational design reduced the bias of the influence of a specific health care system. Using only one language also avoided translation errors. Given that no specific training is necessary to be member of the urogynecologic working groups, we also included non-board-certified gynecologists and had a variety of different levels of experience in our participants. Treatment reimbursement by mandatory health insurances reduced the selection bias of patients who can afford to pay for the treatment.

### 4.8. Limitations

Our survey had several limitations. First, it only investigated BoNT treatment patterns in German-speaking countries. The applicability to other countries might therefore be limited. Second, we could not assess patient outcome data. Consequently, we were not able to analyze the influence of treatment patterns on patient outcomes. Third, we were not able to determine how many active members were contacted because each working group in each country had their own guidelines for how members were allowed to be contacted. All members received an email with a link to the survey, but in one group we were allowed to contact each member personally, while in another group, only mail sent out by the working group office was allowed. Moreover, no complete lists of active and practicing members were available for either urogynecological association. Fourth, gynecologists do not need to be certified urogynecologists to perform urogynecologic procedures in these three countries, and no data base exists on the number of procedures performed. We must assume that certain gynecologists perform this procedure without being members of the urogynecologic society in the corresponding country. Unfortunately, there is no national data base in any of the three countries to verify this.

Our survey confirmed that botulinum toxin is widely used by urogynecologists in the three German-speaking countries, but practice patterns vary widely, and no standardized method could be detected. Board-certified urogynecologists did not apply current guidelines more frequently than non-board-certified gynecologists [8,11]. Furthermore, studies are necessary to investigate the impacts of variations in treatment patterns on patient outcomes. Surveys comparing urologists with urogynecologists are also lacking and might add a meaningful comparison between the two specialties. Our results clearly demonstrate that there is a strong need for standardized perioperative protocols regarding the use of botulinum toxin in patients with OAB.

## Figures and Tables

**Table 1 jcm-12-01462-t001:** Characteristics of respondents. * Cumulative percentage can be more than 100% because participants were allowed to give more than one answer.

Sexual Identification	*n* (%)
Female	49/106 (46%)
Male	57/106 (54%)
Diverse/other	0/106 (0%)
Years in practice	
0–5 years	13/106 (12%)
6–10 years	10/106 (9%)
10–20 years	41/106 (39%)
>20 years	42/106 (40%)
Type of institution *	
University hospital	27/106 (26%)
Non-academic teaching hospital	51/106 (48%)
Non-teaching hospital	22/106 (21%)
Private practice	16/106 (15%)
Board-certified urogynecologist	77/106 (73%)

**Table 2 jcm-12-01462-t002:** Techniques of BoNT application, with comparisons between board-certified urogynecologists and general OBGYNs and between low-volume and high-volume surgeons. Percentages are given as ratios for each subgroup. * Cumulative percentages can be more than 100% because participants were allowed to give more than one answer. Cumulative percentages can be more than 100% because of rounding. BoNT = botulinum toxin, n.s. = not statistically significant, OAB = overactive bladder, OBGYNs = obstetrics and gynecologists. ° Cumulative percentage can be more than 100% because of rounding.

			Board-Certified Urogynecologists (*n* = 77)	General OBGYNs (*n* = 29)	*P-Value*	Low-Volume Surgeons (*n* = 66)	High-Volume Surgeons (*n* = 40)	*P-Value*
Treatment indications								
	idiopathic OAB		94% (72)	83% (24)	n.s.	89% (59)	93% (37)	n.s.
	mixed urinary incontinence with predominant OAB		88% (68)	86% (25)	n.s.	83% (55)	95% (38)	0.026
	neurogenic OAB		67% (51)	69% (20)	n.s.	65% (43)	70% (28)	n.s.
	other indications		8% (6)	10% (3)	n.s.	3% (2)	18% (7)	0.008
Line of treatment *								
	first or second line		14% (11)	3% (1)	n.s.	6% (4)	21% (8)	0.029
	third line		94% (72)	90% (26)	n.s.	89% (59)	95% (38)	n.s.
	fourth line		30% (23)	21% (6)	n.s.	23% (15)	33% (13)	n.s.
Number of transurethral BoNT procedures per year								
	≤20		55% (42)	86% (25)	0.003	-	-	
	>20		45% (35)	14% (4)		-	-	
Drug used *								
	OnabotulinumtoxinA(Botox^®^)		99% (76)	100% (29)	n.s.	99% (65)	100% (40)	n.s.
	AbobotulinumtoxinA(Dysport^®^)		9% (7)	0% (0)	n.s.	3% (2)	13% (5)	n.s.
	IncobotulinumtoxinA(Xeomin^®^)		0% (0)	0% (0)	-	0% (0)	0% (0)	-
Dosage of Botox^®^ at first treatment *								
	50 U		5% (4)	14% (4)	n.s.	6% (4)	11% (4)	n.s.
	100 U		94% (72)	72% (21)	0.001	86% (57)	90% (36)	n.s.
	150 U		4% (3)	10% (3)	n.s.	8% (5)	3% (1)	n.s.
	200 U		6% (4)	10% (3)	n.s.	6% (4)	8% (3)	n.s.
	>200 U		1% (1)	0% (0)	n.s.	0% (0)	3% (1)	n.s.
Dosage of Dysport^®^ at first treatment *								
	<100		0% (0)	0% (0)	-	0% (0)	0% (0)	-
	100		4% (3)	0% (0)	-	2% (1)	5% (2)	n.s.
	200		1% (1)	0% (0)	-	0% (0)	3% (1)	n.s.
	300		3% (2)	0% (0)	-	0% (0)	5% (2)	n.s.
	400		0% (0)	0% (0)	-	0% (0)	0% (0)	-
	>400		3% (2)	3% (1)	n.s.	2% (1)	5% (2)	n.s.
Number of injections °								
	0–10		16% (12)	21% (6)	n.s.	21% (14)	10% (4)	n.s.
	11–20		75% (58)	62% (18)		67% (44)	80% (32)	
	21–30		9% (7)	10% (3)		11% (7)	8% (3)	
	31–40		0% (0)	7% (2)		3% (2)	0% (0)	
Site of injection *								
	Posterior bladder wall		92% (71)	86% (25)	n.s.	91% (60)	90% (36)	n.s.
	Lateral bladder wall		74% (57)	66% (19)	n.s.	70% (46)	75% (30)	n.s.
	Bladder dome		60% (46)	48% (14)	n.s.	62% (41)	48% (19)	n.s.
	Trigone		22% (17)	3% (1)	0.023	6% (4)	35% (14)	<0.001
Usage of a marked needle			69% (53)	59% (17)	n.s.	62% (41)	73% (29)	n.s.
Type of anesthesia *								
	General anesthesia/sedation		34% (37)	55% (16)	n.s.	66% (38)	47% (15)	n.s.
	Local anesthesia		49% (38)	10% (3)	<0.001	27% (18)	58% (23)	0.002
	Spinal anesthesia		28% (17)	54% (15)	0.019	43% (25)	18% (7)	n.s.
	No anesthesia		3% (2)	14% (4)	0.026	5% (3)	8% (3)	n.s.
Perioperative antibiotics			59% (17)	71% (55)	0.015	64% (42)	77% (30)	n.s.
Outpatient vs. inpatient treatment °								
	outpatient		31% (24)	31% (9)	n.s.	33% (22)	28% (11)	n.s.
	inpatient		35% (27)	52% (15)		45% (30)	30% (12)	
	both		34% (26)	17% (5)		23% (15)	41% (16)	

**Table 3 jcm-12-01462-t003:** Postoperative care, with comparisons between board-certified urogynecologists and general OBGYNs and between low-volume and high-volume surgeons. CISC = clean intermittent self-catheterization, n.s. = not statistically significant, PVRV = postvoid residual volume. * Cumulative percentages can be more than 100% because participants were allowed to give more than one answer. ° Cumulative percentage can be more than 100% because of rounding.

			Board-Certified Urogynecologists (*n* = 77)	General OBGYNs (*n* = 29)	*P-Value*	Low-Volume Surgeons (*n* = 66)	High-Volume Surgeons (*n* = 40)	*P-Value*
Postoperative indwelling catheter			48% (37)	48% (14)	n.s.	49% (32)	50% (20)	n.s.
CISC teaching			25% (19)	28% (8)	n.s.	32% (21)	15% (6)	n.s.
Measurement of PVRV *								
	at patient discharge		57% (44)	62% (18)	n.s.	64% (42)	53% (21)	n.s.
	first week		21% (16)	14% (4)	n.s.	18% (12)	23% (9)	n.s.
	first four weeks		47% (36)	31% (9)	n.s.	37% (24)	53% (21)	n.s.
	later		8% (6)	3% (1)	n.s.	8% (5)	5% (2)	n.s.
Repetition of treatment *								
	between 1 and 3 months		9% (7)	7% (2)	n.s.	8% (5)	13% (5)	n.s.
	between 4 and 6 months		31% (24)	28% (8)	n.s.	26% (17)	38% (15)	n.s.
	between 7 and 12 months		38% (29)	31% (9)	n.s.	35% (23)	38% (15)	n.s.
	only after >12 months		3% (2)	0% (0)	n.s.	3% (2)	0% (0)	n.s.
	decision left to the patient		6% (5)	7% (2)	n.s.	8% (5)	5% (2)	n.s.
Maximum number of treatments °					n.s.			
	1		5% (4)	10% (3)	n.s.	8% (5)	5% (2)	n.s.
	2		8% (6)	17% (5)		12% (8)	10% (4)	
	3		16% (12)	10% (3)		20% (13)	5% (2)	
	no limitations		51% (39)	31% (9)		35% (23)	62% (25)	

## Data Availability

The data that support the findings of this study are available from the corresponding author, S.L., upon reasonable request.

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
