# Peer review of "Perioperative Techniques for the Use of Botulinum Toxin in Overactive Bladder: Results of a Multinational Online Survey of Urogynecologists in Germany, Austria, and Switzerland†"

_jcm, 2023, doi:10.3390/jcm12041462_

Round 1

Reviewer 1 Report (Previous Reviewer 2)

The authors provided good responses to my comments, but did not make corresponding changes to the manuscript. In general, responses to reviewers should result in changes to the manuscript so that the readers do not have the same questions.  

Regarding my first question about the number of total physicians who received the survey, the response is valid. However, this explanation should be included in the manuscript as an important limitation. Surveys with lower response rates are less generalizable and more subject to response bias.

Regarding my question about local anesthesia, the authors provide a reasonable response, but they did not make any corresponding changes to the manuscript. The manuscript still recommends that local anesthesia be used as if this article demonstrated superiority of local anesthesia. Please qualify this recommendation and explain that there are no studies demonstrating the superiority of intravesical local anesthesia to placebo in this setting.

Author Response

Response to Reviewer 1 Comments

The authors provided good responses to my comments, but did not make corresponding changes to the manuscript. In general, responses to reviewers should result in changes to the manuscript so that the readers do not have the same questions.  

Regarding my first question about the number of total physicians who received the survey, the response is valid. However, this explanation should be included in the manuscript as an important limitation. Surveys with lower response rates are less generalizable and more subject to response bias.

  • We much appreciated the remark of this reviewer and made the following modifications:
  1. In the section “Material and methods”, we added more information about how we were able to contact participants in the section “participants”.
  2. We added information in the section “limitations of the study” in the discussion part.
  • We sincerely hope that these additions make it clearer for the reader to appreciate the limitations of our study.

Regarding my question about local anesthesia, the authors provide a reasonable response, but they did not make any corresponding changes to the manuscript. The manuscript still recommends that local anesthesia be used as if this article demonstrated superiority of local anesthesia. Please qualify this recommendation and explain that there are no studies demonstrating the superiority of intravesical local anesthesia to placebo in this setting.

  • We would like to thank the reviewer for this comment. In our manuscript we state that local anesthesia is recommended by Taj et al. (Tay LJ, Harry D, Malde S, Sahai A. Cost Effectiveness of Sacral Neuromodulation and OnabotulinumtoxinA in Managing Refractory Idiopathic Overactive Bladder. Urology. 2021 Mar;149:1-10. doi: 10.1016/j.urology.2020.11.018.) to make BoNT more cost effective than sacral neuromodulation. To be clearer, we made modifications to this section.
  • We believe that the reviewer refers to the phrase in line 307-309. We further investigated this matter and found no trial comparing any type of anesthesia with placebo. In a Cochrane analysis by Duthie et al. we found the same recommendation since all included studies used at least on type of anesthesia and we added this citation to the phrase in line 310 (see AUTHORS’ CONCLUSIONS in Duthie JB, Vincent M, Herbison GP, Wilson DI, Wilson D. Botulinum toxin injections for adults with overactive bladder syndrome. Cochrane Database Syst Rev. 2011 Dec 7;(12):CD005493. doi: 10.1002/14651858.CD005493.pub3.). Nevertheless, we added a phrase to the section “Type of anesthesia” concerning the lack of comparison of any anesthesia to placebo in the current literature as demanded by the reviewer (lines 299-300).

Reviewer 2 Report (Previous Reviewer 1)

This research provided some intriguing information of how the BoNT is actually applied by urogynecologists. However, the manuscript still needs careful revision, the description should be rigidly stated, the results should be carefully stated and checked, several data were missing in text, and new tables with further comparisons might be added.

1.          What is the ratio of the person who respond to this survey? Do 122 respondents present evidence strong enough?

2.          Where are the data of “Treatment indications” (line 130 to line 144), “Technique of BoNT application” (line 146) (line 154 to 156) (line 164), and “Postoperative care” (line 167, line 170)? None of the table showed those data.

3.          If the respondents didn’t finish the questionnaire completely, their data should be completely excluded, such situation (n=104/118) (n=96/116) (n=78/117) line 131 to line 134 must be revised.

4.          If authors try to compare the differences of BoNT application between high-volume surgeons and low-volume surgeons, a new table and analysis should be added, rather than merely describe it in text.

5.          Table legend should be placed below table content. Please check the common manuscript guideline.

6.          Please check all of the values in text and figure. Line 148. Author stated “A marked needle to control visually the depth of injection was used by 66% of participants”, but it was 69% in table 2. Line 165 “most participants avoided to inject into this area (75%)”, but the number of who inject into the trigone is 18 (table 2), so the number of who didn’t inject into the trigone is 88, which means the percentage is 83% (88/106). Line 171 “mostly prior to patient’s discharge (71%, see table 3)”, but it is (44+18) / (65+23) = 62/88 (70.4%).

7.          Line 149. “The procedure is performed exclusively under local anesthesia by 16%” Where is this data in table? 41 out of 106 respondents used local anesthesia (in table 2), so where is “16%” from?

8.          Please add the actual number of data when authors state the percentage of each data. For instance, 10% (20/200) of respondents.

9.          Define “OBGYN”.

10.      Line 118. Just “Thirty-seven percent of respondents (n=39/106) declared…”

11.      Some typo could be revised. Line 70, “unclear; how many…”.

Author Response

Response to Reviewer 2 Comments

Comments and Suggestions for Authors

This research provided some intriguing information of how the BoNT is actually applied by urogynecologists. However, the manuscript still needs careful revision, the description should be rigidly stated, the results should be carefully stated and checked, several data were missing in text, and new tables with further comparisons might be added.

  1. What is the ratio of the person who respond to this survey? Do 122 respondents present evidence strong enough?
  • We would like to thank this reviewer for this important comment. As mentioned in our answer to the first remark of Reviewer 1, due to limitations in the way, the three urogynecological associations permitted to contact their members we were not able to determine
  1. How many people received to the survey mail.
  2. How many members of the associations are still active and not i.e., retired or changed specialties, etc.
  • To assure a maximum of transparency about this issue, we added additional information in the sections “material and methos” and “discussion”. We hope that this helps the reader to identify the limitations of our study and shows, which difficulties researchers can encounter when doing surveys on multinational levels.

  1. Where are the data of “Treatment indications” (line 130 to line 144), “Technique of BoNT application” (line 146) (line 154 to 156) (line 164), and “Postoperative care” (line 167, line 170)? None of the table showed those data.
  • We appreciate this reviewer’s comment, and we added information to the table for the mentioned sections.
  1. If the respondents didn’t finish the questionnaire completely, their data should be completely excluded, such situation (n=104/118) (n=96/116) (n=78/117) line 131 to line 134 must be revised.
  • We agree with the reviewer that incomplete questionnaires should be excluded, hence we corrected this. Additionally, we corrected our methods section to explain how we treated incomplete surveys.
  1. If authors try to compare the differences of BoNT application between high-volume surgeons and low-volume surgeons, a new table and analysis should be added, rather than merely describe it in text.
  • We thank the reviewer for this comment and according to the demand, we mode the following changes:

- Table 2 and 3: we added columns for high- and low-volume surgeons

- We modified text passages discussing results of these groups.

  1. Table legend should be placed below table content. Please check the common manuscript guideline.
  • We made the demanded modifications.
  1. Please check all of the values in text and figure. Line 148. Author stated “A marked needle to control visually the depth of injection was used by 66% of participants”, but it was 69% in table 2. Line 165 “most participants avoided to inject into this area (75%)”, but the number of who inject into the trigone is 18 (table 2), so the number of who didn’t inject into the trigone is 88, which means the percentage is 83% (88/106). Line 171 “mostly prior to patient’s discharge (71%, see table 3)”, but it is (44+18) / (65+23) = 62/88 (70.4%).
  • We checked the values as demanded:
    • “A marked needle to control visually the depth of injection was used by 66% of participants”, but it was 69% in table 2.
      • The 69% concerns the number of board-certified urogynecologists (n=53/77) who used a marked needle to inject BoTN. The 66% indicate the total number of participants in the survey who used marked needles (n=70/106).
      • To avoid confusion, we added an explanation to the mentioned section.
    • Effectively, the number 75% was erroneous, and we made corrected this error.
    • There seemed to be a rounding error because we rounded from 70.45% to 70.5% and then to 71%. Nevertheless, we made modifications to this section because in our first analysis we did not include participants who did not measure PVRV at all but only those who measured at a certain timepoint. Now, we included these participants in the group who did not perform the measurement at the demanded timepoint.
      • Example: in the first analysis, 18 patients did not measure PVRV at all and therefore, only 88 participants were counted, resulting in 70.4% of participants who measured PVRV at discharge (62/88). Now, all 106 participants are counted. This results in 59% (rounded from 58.5%).
  1. Line 149. “The procedure is performed exclusively under local anesthesia by 16%” Where is this data in table? 41 out of 106 respondents used local anesthesia (in table 2), so where is “16%” from? 
  • It is true that we did not mention that 16% of participants stated only local anesthesia as type of anesthesia. We agree with the reviewer that our wording was confusing, and we therefore corrected this section.
  1. Please add the actual number of data when authors state the percentage of each data. For instance, 10% (20/200) of respondents.
  • As demanded, we added the actual number of data wherever missing.
  1. Define “OBGYN”.
  • We added a definition of OBGYN to the subsection “Participants” in the “Material and methods” section.
  1. Line 118. Just “Thirty-seven percent of respondents (n=39/106) declared…”
  • We made the proposed modifications.
  1. Some typo could be revised. Line 70, “unclear; how many…”.
  • We corrected the typo.

Round 2

Reviewer 2 Report (Previous Reviewer 1)

The overall issues are addressed by authors and the manuscript is comprehensible and transparent for readership. A few amendments could be further achieved.

1.          CISC should be defined in abstract.

2.          Please check whether the place of p value (0.003) of the analysis of “Number of transurethral BoNT procedures per year” (table 2) was correct, it was placed between two cells.

3.          Typo in line 389 (determine”,” how many active members).

Author Response

Dear reviewer, we appreciated the comments and made corrections where necessary. You can find our answers below:

  1. CISC should be defined in abstract.
    • We corrected this and defined CISC in the abstract.
  2. Please check whether the place of p value (0.003) of the analysis of “Number of transurethral BoNT procedures per year” (table 2) was correct, it was placed between two cells.
    • We placed the p-value on purpose between the two lines given that it indicates the result of the chi-square test of the two groups concerning the number of participants who perform more or less than 20 procedures per year. To be transparent we want to state that we proceeded in the same manner in the same table for "number of injections" and in table 3 for "number of maximum treatments". This was different to other questions (i.e., "treatment indications") because the questions were of the single-answer type and therefore, comparisons between the groups had to be done for all possible answers and not for each individual answer as it was done when multiple answers per question were possible. We hope that our reply satisfies the reviewer.
  3. Typo in line 389 (determine”,” how many active members).
    • We corrected the typo.

Additionally, we would like to inform the reviewer that we made slight changes to the first line of the tables 2 and 3, where we replaced "board-certification" with "board-certified urogynecologists". We think that this makes it easier and quicker for the readers to understand the table.

This manuscript is a resubmission of an earlier submission. The following is a list of the peer review reports and author responses from that submission.

Round 1

Reviewer 1 Report

This survey proposes the clinicians’ opinion of the use botulinum toxin for treating overactive-bladder. The results reveal the diverse clinical practice of intravesical botox injection and suggest that the standardized protocol should be further established. This research has some merits and proposes some interesting ideas. However, thorough amendments and supplements should be carried out. We suggest resubmission after thorough revision. Authors should at least revise the following issues, as follows:

1. The guidelines of botulinum toxin injection for bladder disease is clear and well-established, it is not unclear as authors declared. These introduction should be objective and scientifically evidence-based. The references that authors cited also state the guidelines and suggestions of botulinum toxin. Authors may refer to American Urological Association or European Association of Urology guidelines and related research (https://doi.org/10.1016/j.euf.2018.10.011) (https://doi.org/10.1111/iju.14176) (https://doi.org/10.1016/j.eururo.2022.03.010) (https://doi.org/10.1097/JU.0000000000002709).

2. The mechanism of botulinum toxin injection to treat overactive-bladder should be supplemented. Authors may refer to the following reference (https://doi.org/10.1016/j.juro.2016.11.092).

3. The current pharmacologic treatments for OAB or neurogenic bladder dysfunction should be mentioned in introduction or discussed. Suggested reference: (https://doi.org/10.1016/j.eururo.2020.12.032) (https://doi.org/10.1097/JU.0000000000002239).

4. Intravesical drug instillation, including botulinum toxin and other materials, for treating bladder disorder should be mentioned or discussed, the suggested references are as follows: (https://doi.org/10.3390/ph14050409) (https://doi.org/10.1016/j.ijpx.2021.100100).

5. Several data described in the whole section of result were not presented in any table. For instance, “but only three-quarter (77%, n=92/122) of these” in line 103, “Slightly more than one-third of participants (37%) declared themselves” in line 105, “non-board-certified gynecologists (46% vs 10%, p = 0.003)” in line 107, “with a predominant OAB (91% and 89%, resp.)” in line 118, etc. Please check all of the data written in section of results and add the data in table.

6. The number of participants responded to survey is 122. So, why the total number of participants in table is 106? Please check all of the values in manuscript and tables.

7. The description such as “three-quarter”, “one-third of participants”, “Nine out of ten participants” should not be used, just use the actual number or percentage, such as “77% of the participants”.

8. We are not sure if the format and the reference style are accordingly applied or based on journal’s submission guideline, please check again.

Reviewer 2 Report

The authors present a survey study on practice patterns for intra-vesical Botulinum Toxin injection. They found no standardized protocols between providers. The study is well conducted, and the manuscript is well written. It might be improved by considering the following comments.

Results: How many total physicians received the survey. It is essential to know this denominator to determine the survey response rate.

Discussion: lines 166-167: This sentence is worded poorly and difficult to understand. Consider rewriting.

Discussion lines 226-227: The authors recommend that local anesthesia be used. However, this study is not at all designed to make such a recommendation. Moreover, there is significant variability in the way local anesthesia can be administered to the bladder in terms of the concentration, volume, and instillation time. What is the evidence that local anesthetic is effective in reducing pain from Botox injection compared to no anesthetic or placebo?

Survey studies provide no data to elucidate the risks and benefits of the various protocols; they just show that there is a high degree of variation between providers. Therefore, any data from a survey study should not be used to guide clinical practice. This study mainly serves to justify the need for clinical trials in this space.